# Time trajectories in the transcriptomic response to exercise - a meta-analysis

David Amar [1,4], Malene E. Lindholm [1,4], Jessica Norrbom[2], Matthew T. Wheeler [1], Manuel A. Rivas [3] & Euan A. Ashley [1✉]

Exercise training prevents multiple diseases, yet the molecular mechanisms that drive exercise adaptation are incompletely understood. To address this, we create a computational framework comprising data from skeletal muscle or blood from 43 studies, including 739 individuals before and after exercise or training. Using linear mixed effects meta-regression, we detect specific time patterns and regulatory modulators of the exercise response. Acute and long-term responses are transcriptionally distinct and we identify *SMAD3* as a central regulator of the exercise response. Exercise induces a more pronounced inflammatory response in skeletal muscle of older individuals and our models reveal multiple sex-associated responses. We validate seven of our top genes in a separate human cohort. In this work, we provide a powerful resource (www.extrameta.org) that expands the transcriptional landscape of exercise adaptation by extending previously known responses and their regulatory networks, and identifying novel modality-, time-, age-, and sex-associated changes.

[1] Center for Inherited Cardiovascular Disease, School of Medicine, Stanford University, Stanford, CA, USA. [2] Department of Physiology & Pharmacology, Karolinska Institutet, Stockholm, Sweden. [3] Department of Biomedical Data Science, School of Medicine, Stanford University, Stanford, CA, USA. [4] These authors contributed equally: David Amar, Malene E. Lindholm. ✉email: euan@stanford.edu

Performing regular exercise is one of the most important actions by which individuals of all ages can improve their health. It prevents multi-system chronic diseases, reduces anxiety, improves cognitive function, and overall quality of life[1,2]. Recent studies applying mendelian randomization have also established the causal benefit of exercise in the prevention of depression[3], bipolar disorder[4] and breast and colorectal cancer[5], while no effect was observed for Parkinson's disease[6] or schizophrenia[4]. Both acute effects and long-term adaptation have been studied in response to endurance (repeated contractions of low force) and resistance (fewer contractions of greater force) exercise in humans, where skeletal muscle and blood are the most accessible and well-studied tissues. Exercise-induced adaptation occurs in working skeletal muscles, however, multiple myokines are released from skeletal muscle into blood, with the potential to affect all organs[7]. Resistance training specifically leads to skeletal muscle hypertrophy, while endurance training leads to increased capillary density, ATP-producing capacity and improves cardiorespiratory fitness[8]. Both exercise modalities improve insulin sensitivity[9], decrease blood pressure[10], and improve the blood lipid profile by increasing high-density lipoprotein cholesterol and reducing low-density lipoprotein and triglyceride levels[11–13]. Several key transcriptional pathways in skeletal muscle are also common between the two exercise modalities[14].

The global transcriptional response to both acute exercise and long-term training, measured using microarrays or RNA-sequencing, has been investigated in healthy humans by sampling human blood[15–19] and skeletal muscle[20–25]. These studies identified multiple differentially expressed genes. However, current studies of the molecular response to exercise are limited in size due to the complexity of conducting controlled interventions in healthy individuals that include invasive biological sampling, something that the NIH common fund initiative the Molecular Transducers of Physical Activity is aiming to address[26]. Consequently, available datasets differ substantially in the clinical attributes of their subject sets, including: sex, age, training modality, and sampled post-exercise time points.

Meta-analysis is a standard tool for systematic quantitative analysis of previous research studies. It can generate precise estimates of effect sizes and is often more powerful than any individual study contributing to the pooled analysis[27]. Moreover, the examination of variability or heterogeneity in study results is also critical, especially when apparently conflicting results appear. Meta-analysis can be based on simple weighted averaging using random-effects (RE) models, or, in more complex cases, be based on meta-regression of effect sizes vs. covariates, which are typically assumed to be effect moderators. Meta-analysis (and meta-regression) has been traditionally used in medicine and epidemiology, but it has been gaining popularity in large-scale genomics studies[28,29]. Recently, Pillon et al.[30] presented a random-effects meta-analysis of exercise gene expression studies from skeletal muscle. The analysis was split based on the training modality and type (e.g., acute endurance exercise studies were analyzed separately from other studies), and thousands of genes were identified to have differential expression, with NR4A3 (nuclear receptor subfamily 4 group A member 3) as the top candidate gene. However, the analysis accounted for only a single moderator, and the dependence of the effects on additional moderators was not analyzed systematically. Thus, the dynamic response of genes over time was not inspected in a systematic way for all genes.

In this work, we gathered and annotated publicly available transcriptome datasets from both endurance and resistance exercise interventions in humans, covering 1724 samples from human blood or skeletal muscle from 739 subjects in total. We address the statistical issues discussed above by (1) fitting a model for each gene while accounting for the explained variability by moderators including: sex, age, training type, and time post exercise, (2) applying a set of filters on the results to promote replicability, as suggested in ref. [28], and (3) integrating the meta-analysis results with various biological networks using systems biology methods. Our results expand the current understanding of the transcriptional landscape of exercise adaptation by extending previously known expression responses and their regulatory networks, and identifying modality-, time-, age-, and sex-associated changes. We identify SMAD3 (SMAD family member 3) as a central regulator of the acute exercise response, observe a more pronounced inflammatory response to training in older individuals and find multiple sex-associated differentially regulated genes, where MTMR3 (myotubularin related protein 3) has not been associated with exercise before.

## Results

**Data collection.** We collected and annotated data from 43 human exercise training studies, including 1724 samples from human blood or skeletal muscle from 739 subjects in total (see Methods). Figure 1 shows an overview of the study. Dataset records from the Gene Expression Omnibus (GEO), manuscripts, Supplementary Information, and personal communication with authors were used to extract sample-level or study-level information including sex, age, tissue, training modality, additional treatments, and time points. Transcriptome data from these samples were used for the meta-analysis. After imputing missing sex information in 116 subjects using the expression of the Y-chromosome genes (internal validation was at 0.99 receiver operating characteristic (ROC), see Methods), 310 of the 739 subjects were female, 409 were males, 18 had no sex information, and two subjects had discordant sex predictions. Each dataset was partitioned into homogeneous cohorts that had a similar training regime, see Supplementary Data 1 for details. This resulted in 59 cohorts, 13 for blood and 46 for skeletal muscle. For each gene and post-exercise time point, we computed the effect size of differential expression compared to the pre-exercise sample in each study. In addition, for each cohort we kept the following four *moderators* (covariates): average age, proportion of males, time points, and training modality. Additional potential effect moderators including (but not limited to) height, weight, body mass index (BMI), training status, baseline heart rate, maximal oxygen uptake ($VO_2max$), and exact location of muscle biopsy were all examined but were excluded because of low coverage. For example, only 188 samples (out of 1724) had weight information. Nevertheless, we provide these variables in our curated dataset (see Supplementary Data 2–3).

**Meta-analysis discovers multiple pathways.** We partitioned the analysis by tissue (blood or muscle) and intervention (acute exercise bout vs. long-term training). For each type we ran a simple random-effects (RE) meta-analysis using all summary statistics, but with adjustment for study-level random effects. We then ranked all genes by their inferred fold-change, and ran gene set enrichment analysis (GSEA) for pathway enrichment[31]. This analysis resulted in multiple enriched pathways in each of our four analyses. Figure 2a shows the four-way Venn diagram of the pathway sets discovered in the analyses at 10% Benjamini–Yekutieli false-discovery rate (FDR) correction[32]. The two pathways that appear as significant in all four analyses are related to respiratory electron transport. Supplementary Fig. 1 presents the top three up- and three downregulated pathways in each analysis, see Supplementary Data 4 for all significant GSEA results. These results illustrate how the meta-analysis can detect relevant pathways in each analysis, but some pathways appear

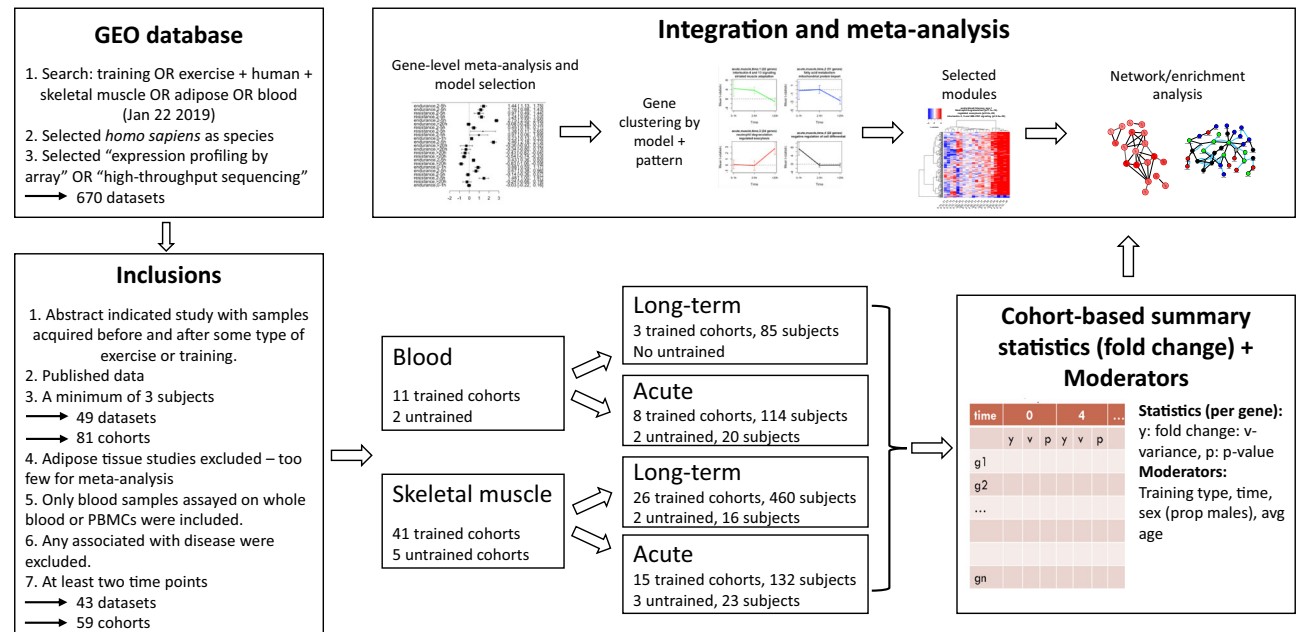

**Fig. 1 Study overview.** We started with a search in the Gene Expression Omnibus. Manual examination of the studies and a set of filters resulted in 43 studies that had whole-genome expression profiles from blood and muscle. The data covered 59 exercise cohorts that were partitioned into four types of meta-analysis (i.e., by tissue and exercise modality). Seven-hundred and thirty-nine subjects were included in total, where some are represented in several cohorts, for example when sampling was made in association to both acute exercise and long-term training.

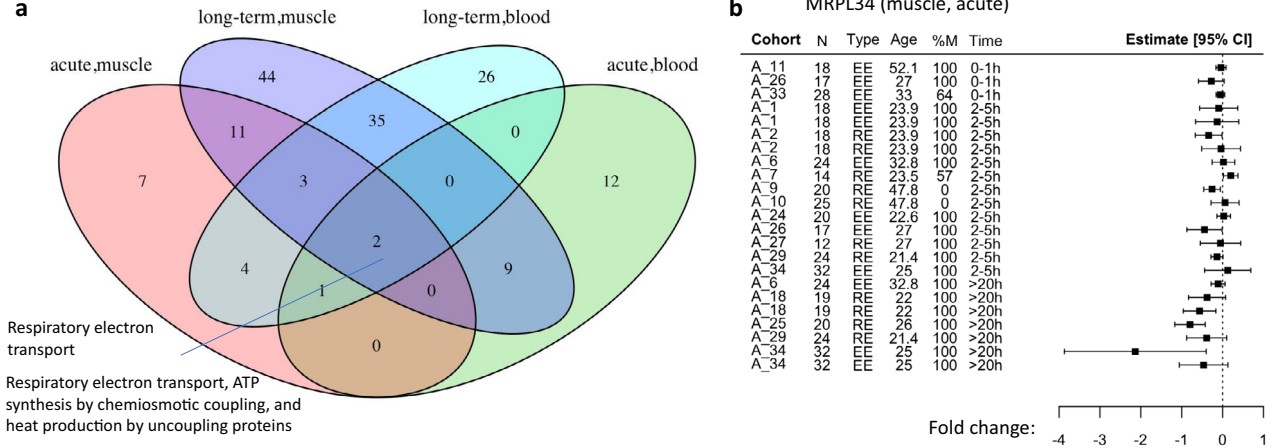

**Fig. 2 Random-effects meta-analysis results. a** Venn diagram of the significant GSEA pathway sets discovered with 10% Benjamini–Yekutieli FDR correction. The two common pathways are listed on the left. Pink—acute muscle changes, purple—long-term muscle changes, turquoise—long-term blood changes, green—acute blood changes. **b** Forest plot of the effect sizes of *MRPL34* in acute muscle cohorts illustrates the importance of adding time as a moderator. Rows represent the 95% confidence interval of a fold-change of a cohort in a given time point. Thus, in each interval the center represents the fold-change estimate and the error bars are proportional to the fold-change standard error. Downregulation is observed only in time points >20 h. Cohort: the ID given to the cohort in this study (see Supplementary Data 1 for details), N: sample size, Type: exercise type, RE: resistance exercise, EE: endurance exercise, Age: mean age in cohort, %M: percent of males in cohort, Time: the time window in hours. Note that a cohort can have multiple rows with the same time window and different fold-changes (e.g., if a study measured both 2 h and 4 h then both time points will be assigned 2–5 h). Source data are provided as a Source Data file.

with reversed direction of effect, which is expected when important covariates are ignored[33]. For example, mitochondrial translation has a negative normalized enrichment score (NES) in the acute muscle meta-analysis (NES −2.17). When examining the top ranked gene in this pathway, *MRPL34* (mitochondrial ribosomal protein L34), we observe that in most cohorts it is not differentially expressed, and it is downregulated only in time points >20 h (Fig. 2b).

The *MRPL34* example illustrates a case with high heterogeneity of the effect sizes ($I^2 = 65\%$). When such heterogeneity is high and the number of studies is moderate or low (as in our case), statistical analyses may be unreliable, resulting in potential inflation of false positives. Indeed, we observe that thousands of genes in each analysis have excess heterogeneity ($I^2 > 70\%$), see Supplementary Fig. 2a. Moreover, we observe an inflation of low *p*-values of the meta-analysis models, see Supplementary Fig. 2b.

As observed in previous studies, such results may lead to low replicability even of the top ranked meta-analysis genes[28,34]. To summarize this section, we observe that a standard naive meta-analysis results in many detected pathways, but inference at the single gene level may result in high error rate, which is in line with previous observations in other domains.

**Model selection for single gene analysis**. The clinical heterogeneity (e.g., variation in age, sex distribution, baseline fitness and type of exercise intervention) in our dataset stems from covering different populations, and unbalanced sampling of the moderators. For example, the acute studies tended to cover different post-exercise time points, see Supplementary Data 1. When examining the associations between the potential moderators across the muscle cohorts, we observed generally mild associations (e.g., median $R^2 < 12\%$, insignificant $p$-values), with the exception of a marked bias of long-term muscle studies towards young males, see Supplementary Fig 3.

To mitigate the issues discussed above we developed a pipeline that fits a model for each gene (see Methods). Briefly, we considered all combinations of the represented moderators, each producing a different meta-regression model. The naive model without any moderators from the previous section is denoted as the *base model*. Model selection was performed using the Akaike information criterion with correction for small sample sizes (AICc)[35,36]. However, to focus on robust genes, we applied an additional series of conservative filters, including both a non-negligible fold-change (>0.1), and a high significance of the top model ($p < 1 \times 10^{-4}$, corresponds to FDR < 0.1 over all tests). Moreover, a model was considered a significant improvement over the base model only if the AICc improvement was >5. Otherwise, we took the base model, but only if the inferred heterogeneity was reasonably low ($I^2 < 50\%$).

We detected differential genes in each of the four analyses: acute exercise, muscle: 537 genes; long-term training, muscle: 441 genes; acute exercise, blood: 37 genes; and long-term training, blood: 48 genes. The low number of genes from the blood analyses is due to several factors: (1) lower number of cohorts, (2) limited statistical power (e.g., only 8650 genes had at least two $p$-values < 0.05 in the acute datasets as compared to 13,016 in muscle), and (3) lower effect sizes (e.g., only 470 genes had fold-change >0.1 in the acute,blood base models).

We further clustered each gene set into co-expressed gene groups (see Methods). Supplementary Data 5–9 provide the statistics of the selected genes, and Supplementary Data 10–11 provide their Gene Ontology (GO) and pathway enrichment results (at 10% FDR adjustment). Note that as the blood datasets were limited in their numbers and coverage, we did not include moderators (i.e., genes were selected using the base models only).

To illustrate the output of the algorithm, consider the response of *PPARGC1A* (PPARG coactivator 1 alpha or PGC-1α) in skeletal muscle after acute exercise. This gene is an exercise-mediated regulator of muscle adaptation with a well-documented transcriptomic response[37]. In our analysis, *PPARGC1A* passed all statistical tests above and its top model included time as a moderator. Its forest plot (Fig. 3a) shows a consistent upregulation at the intermediate (2–5 h) time point following exercise, with a mild downregulation at later time points (>20 h). As another example, consider *COL4A1* (collagen type IV alpha 1 chain), a type IV collagen gene, which showed a consistent induction with long-term training across studies (Fig. 3b).

In muscle, most detected genes had one or more moderators in their selected model, see Supplementary Fig. 4 for co-occurrence of covariates in the selected models. Note that as we tested for both linear and quadratic trends over time, these patterns were

selected together in many cases. An example is shown in Supplementary Fig. 4c, where linear or quadratic patterns alone cannot fully capture the time-associated pattern in which the gene is upregulated at the 0–1 h window but has no differential expression in the next time windows.

To further corroborate our approach, we performed a meta-analysis of the untrained cohorts, which were ignored in the main analysis above. We compared the inferred effect sizes of our selected genes in the exercise cohorts with the untrained cohorts, see Fig. 3c, showing that the effects were exercise-specific.

The acute response in blood was mainly associated with immune regulation. Figure 3d shows the heatmap of the upregulated acute blood genes, where the top enrichment was neutrophil degranulation. In response to long-term training in blood, downregulated genes were enriched for glycerophospholipid biosynthesis, and upregulated genes enriched for peptide chain elongation (Supplementary Data 10–11).

**Temporal patterns imply information propagation across gene networks**. We identified 159 genes in the acute muscle meta-analysis with only time as the selected moderator. Our clustering analysis partitioned this set into four groups based on their time courses, see Fig. 4a (see Methods for details about the clustering algorithm). Each cluster corresponds to a different time trajectory and a different functional annotation. A large connected component with representation from all clusters appears when overlaying the genes on known pathway or protein–protein interactions (PPI), see Fig. 4b. The main hub of the network is *SMAD3*, which is an early-mid upregulated gene. It is an intracellular effector of TGF-β, involved in regulating the balance between protein synthesis and degradation. *SMAD3* is more active in obese individuals[38], whereas lacking *SMAD3* protects mice against diet-induced obesity and insulin resistance[39]. *SMAD3* is required for the atrophic effect of myostatin[40], and has shown a greater induction in response to resistance training in females compared to males[41]. Having *SMAD3* as the main hub together with other early upregulated genes as first and second degree neighbors, suggests that their subnetwork responds first and its effect is then propagated downstream, providing novel information about the early transducers of the transcriptomic response to exercise.

An early induced exercise response gene was *HES1* (Hes family BHLH transcription factor 1), a basic helix-loop-helix repressor. It is a Notch target gene involved in skeletal muscle differentiation[42], but its regulatory role in exercise remains to be elucidated. Another hub was *SH3KBP1* (SH3 domain-containing kinase binding protein 1), which was identified in the late downregulated cluster. Our analysis also identified many known exercise-responsive genes, including PGC-1α, a central regulator of mitochondrial biogenesis that is well-studied in relation to exercise adaptation[37,43]. Genes related to angiogenesis were also induced early, for example *PDGFB* (platelet-derived growth factor B) and *VEGFA* (vascular endothelial growth factor A) that belong to the same protein family. *SPP1* (secreted phosphoprotein 1) is an example of a gene that increased in the late time window (>20 h post exercise) (Fig. 4a, b). Its expression is stimulated by PGC-1α, and its protein is subsequently secreted to activate macrophages and induce angiogenesis[44]. Genes related to fatty-acid metabolism were downregulated during the later time points following acute exercise (Fig. 4a, late down cluster). Examples of genes with this expression pattern are *MLYCD* (malonyl-CoA decarboxylase), which produces Acetyl-CoA from Malonyl-CoA and thus stimulates fatty-acid oxidation; and *CPT1B* (carnitine palmitoyltransferase 1B), which is a mitochondrial membrane transporter of fatty acids.

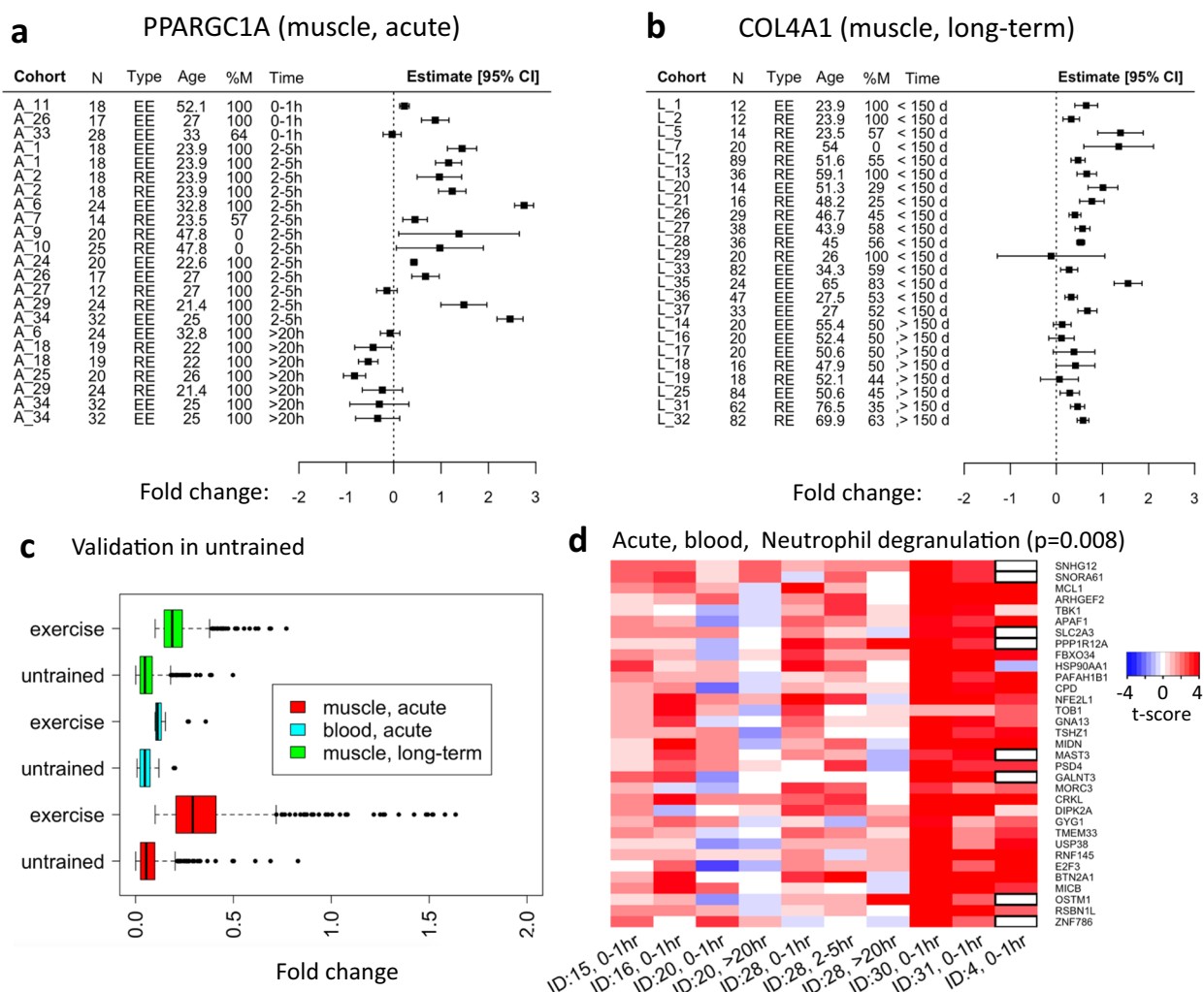

**Fig. 3 Properties of the meta-analysis results. a** Forest plot of the effect sizes of *PPARGC1A* in acute muscle cohorts. Upregulation appears primarily in the 2–5 h time window and is consistent for both endurance and resistance training. **b** Forest plot of the effect sizes of *COL4A1* in the long-term muscle cohorts. Upregulation appears consistently across the cohorts. **a–b** Rows represent the 95% confidence interval of a fold-change of a cohort in a given time point. Thus, in each interval the center represents the fold-change estimate and the error bars are proportional to the fold-change standard error. Cohort: the ID given to the cohort in this study (see Supplementary Data 1 for details), N: sample size, Type: exercise type, RE: resistance exercise, EE: endurance exercise, Age: mean age in cohort, %M: percent of males in cohort, Time: the time window in hours. Note that a cohort can have multiple rows with the same time window and different fold-changes (e.g., if a study measured both 2 h and 4 h then both time points will be assigned 2–5 h). **c** The differential expression effects of our selected genes is unique to exercise. Colors represent the different meta-analysis types in the study. For each meta-analysis the effect sizes of our selected genes are presented, once for the exercise cohorts and once for the untrained cohorts. All paired two-sided Wilcoxon rank sum tests were significant at $p < 1 \times 10^{-07}$ (acute exercise, muscle: 537 genes, $p = 1.5 \times 10^{-88}$; long-term training, muscle: 441 genes, $3.1 \times 10^{-68}$; acute exercise, blood: 37 genes, $p = 6.2 \times 10^{-08}$). Each boxplot shows the median, and first and third quartiles. The whiskers extend from the hinge to the largest and lowest values, but no further than 1.5 *(the inter-quantile range). **d** Heatmap of the upregulated genes selected in the acute blood meta-analysis. Columns represent cohorts, with sampling time point post exercise listed for each, and rows are genes. Boxed cells indicate missing values for that particular gene and cohort. All acute blood studies measured the response to endurance exercise in up to 3 h after the bout ended. Source data are provided as a Source Data file.

We used the dynamic regulatory events miner (DREM)[45] to integrate the time courses with known transcription factor-target networks and sequence information. DREM produces a model that links temporal responses to putative regulators. In our data, DREM predicted four transcription factors to be important upstream transcriptional regulators during the early 0–1 h time window (Fig. 4c), where the top two were BHLHE40 (basic helix-loop-helix family member E40) and HDAC2 (histone deacetylase 2). BHLHE40 is induced by hypoxia and directly binds to PGC-1α to repress its transactivation activity, likely to reduce hypoxia-induced reactive oxygen species damage. The repression of PGC-1α by BHLHE40 has been shown to be alleviated by an increase in *PPARGC1A* expression itself in vitro, and by running exercise in

mice[46]. Moreover, there is evidence that *BHLHE40* is itself induced by a bout of exercise in human skeletal muscle[47]. HDAC2 is a class I histone deacetylase whose role in exercise is largely unknown. Interestingly, lower HDAC2 has been suggested to cause skeletal muscle weakness in patients with chronic obstructive pulmonary disease[48]. The two other identified factors were NR3C2 (nuclear receptor subfamily 3 group C member 2) and RDBP (or negative elongation factor E encoded by the *NELFE* gene), which have not been previously associated with regulation of the exercise response.

**Replication of acute exercise-regulated genes.** We used a separate human acute exercise cohort to validate the main genes with

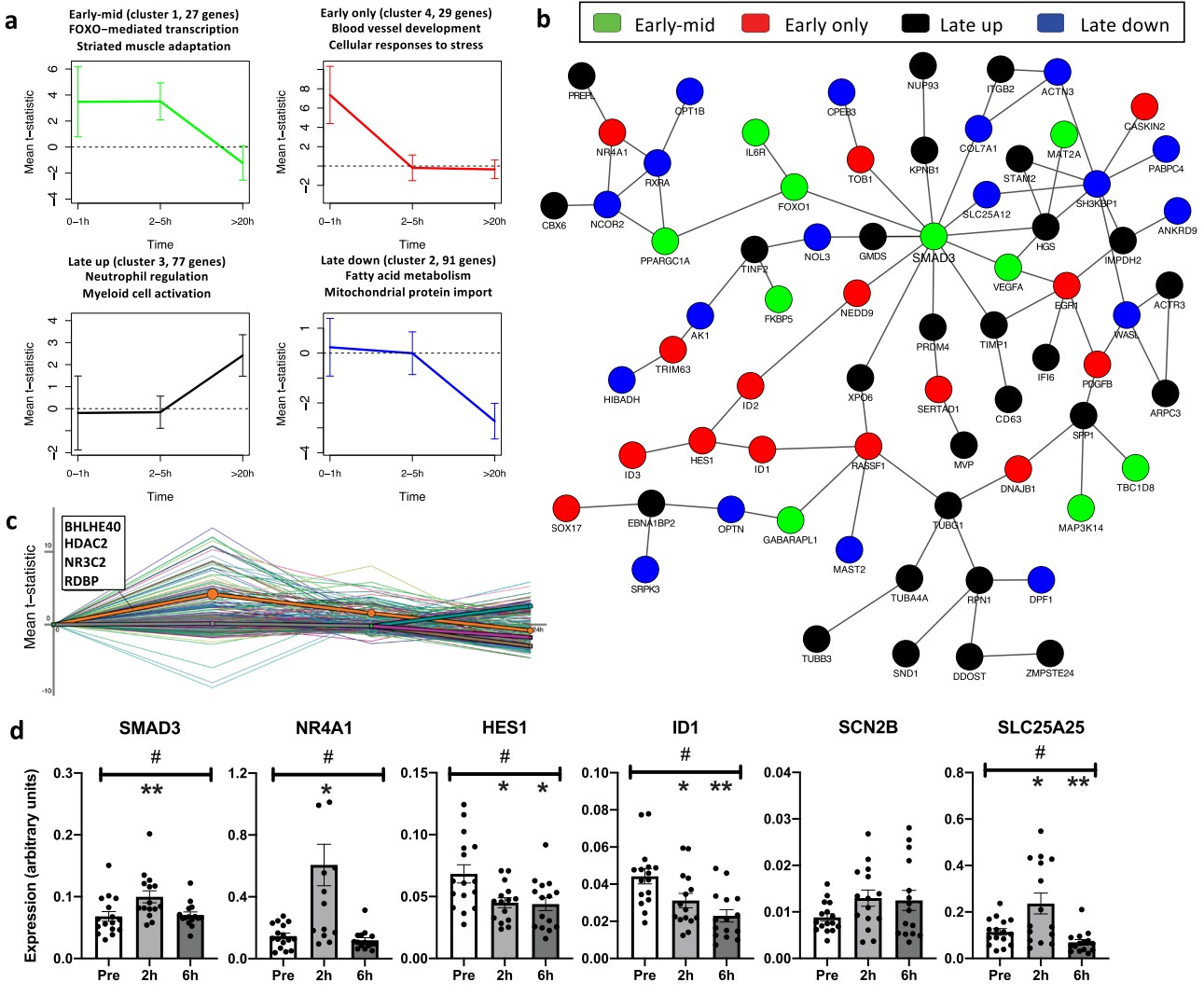

**Fig. 4 Differential expression patterns in skeletal muscle after acute exercise. a** Genes associated with time were partitioned into four groups based on their trajectories. The title of each subplot shows significantly enriched Gene Ontology terms or pathways. **b** The main GeneMANIA connected component of the genes in **a** when overlaid on known protein–protein or pathway networks. **c** Dynamic regulatory events miner (DREM) analysis results of the genes in A predict several transcription factors to be involved in the observed responses. **d** Validation of gene expression changes following acute endurance exercise in a separate human cohort ($n = 16$). *SMAD3, NR4A1, HES1* and *ID1* from the acute network and *SCN2B* and *SLC25A25* from the endurance and time-specific genes. Expression levels are calculated relative to the average of two housekeeping genes (*GAPDH* and *RPS18*). # denotes an overall significant treatment effect between time points ($p < 0.05$). *($p < 0.05$) and **($p < 0.005$) denote significant expression difference compared to before exercise (pre) after correction for multiple comparisons. Values are presented as mean±standard error of the mean. Source data are provided as a Source Data file.

quantitative real-time PCR (qRT-PCR) (Fig. 4d). Sixteen subjects, 8 males and 8 females, performed 1 h of endurance exercise at 70% of their peak VO₂. Skeletal muscle biopsies were obtained before, and at 2 h and 6 h after exercise. *SMAD3* and *NR4A1* were validated as part of the central acute cluster network, and in accordance with the meta-analysis findings, both were significantly upregulated at the 2 h time point following exercise. *HES1* and *ID1* (inhibitor of DNA binding 1) were induced in the early 0–1 h window in the meta-analysis, which was not covered in the validation cohort. However, both genes showed significant positive quadratic, and negative linear trends in the meta-analysis. Supplementary Fig. 5 illustrates how these genes are mostly downregulated in the 2–5 h window. Our validation showed a significant downregulation at both 2 h and 6 h after exercise, which is in agreement with the meta-analysis. *SCN2B* (sodium voltage-gated channel beta subunit 2) was upregulated, although not part of the main network, and *SLC25A25* (solute carrier

family 25 member 25) was differentially regulated in response to both acute exercise and long-term training. Both showed significant upregulation in the early time windows. While upregulated in both tested time points, the change of *SCN2B* was not significant in the validation cohort ($p = 0.09$), while *SLC25A25* significantly increased at the 2 h time point, and significantly decreased at 6 h, in concordance with the >20 h time window for the meta-analysis.

**Training induces extracellular matrix genes in skeletal muscle**. We identified ten downregulated and 104 upregulated genes in the skeletal muscle, long-term training meta-analysis (Fig. 5 and Supplementary Data 8). Myostatin (*MSTN*), which is well-characterized as a negative regulator of skeletal muscle hypertrophy, was one of the downregulated genes. The upregulated genes were enriched for extracellular matrix (ECM) reorganization and laminin interactions, a central training adaptation

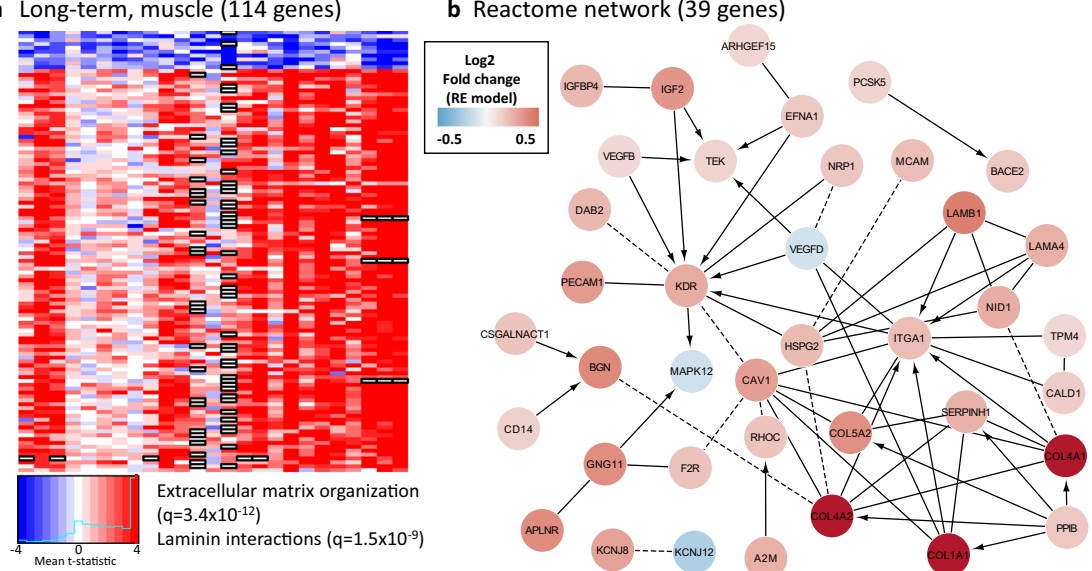

**Fig. 5 Long-term training muscle response genes with moderator-independent patterns. a** Heatmap of the ten downregulated and 104 upregulated genes **b** The main connected component linking most of the upregulated genes to known protein–protein interaction networks. Highlighted nodes are genes with known pathway interactions in Reactome. The laminin interactions Reactome pathway covers only a small part of the upregulated gene module. Dashed lines represent co-occurrence in a complex. Directed edges represent regulation. Source data are provided as a Source Data file.

mechanism[23,49]. A representative reactome network is shown in Fig. 5b. Several collagen genes important for ECM, including *COL4A1*, *COL4A2* (collagen type IV alpha 2 chain), *COL1A1* (collagen type I alpha 1 chain) and *COL5A2* (collagen type V alpha 2 chain) were highly induced. *COL4A1*, for example, represents one of the major components of the basement membrane surrounding skeletal muscle fibers[50]. A cell surface receptor for collagens and laminins, *ITGA1* (integrin subunit alpha 1), was identified as one of the central regulatory hubs. Another central regulator was *KDR* (kinase insert domain receptor or VEGF receptor 2) indicating the key role of angiogenesis for training adaptation in skeletal muscle.

Although repeated bouts of acute exercise over a longer period of time evidently lead to adaptation, the gene-level transcriptional overlap between acute and long-term exercise responses was low (Supplementary Fig. 6), which is similar to previous observations for resistance exercise and training[14]. Only 13 genes were differentially expressed in response to both acute exercise and long-term training in skeletal muscle, including the mitochondrial carrier *SLC25A25*, the potassium channel *KCNC4* (potassium voltage-gated channel subfamily C member 4) and the malonyl-CoA decarboxylase *MYLCD*.

**More pronounced inflammatory response to exercise with age.** The sample size, and heterogeneity of the cohorts regarding sex and age allowed us to interrogate potential moderator-specific transcriptional alterations in response to acute exercise and long-term training in the skeletal muscle cohorts.

Differential expression of 76 genes was significantly associated with age in the acute exercise response (Supplementary Data 6). Figure 5a shows the genes associated with age and time post exercise, where notably the nuclear receptors *NR4A2* (nuclear receptor subfamily 4 group A member 2) and *NR4A3* are represented. *NR4A3* was recently highlighted to be oppositely regulated in exercise and inactivity, and silencing of *NR4A3* decreased oxygen consumption in skeletal muscle cells in vitro[30]. In response to long-term training, 73 genes were significantly associated with age. Particularly, for time and age, we found that older individuals showed greater induction of interferon-induced

inflammatory genes, including *HLA-DRA* (major histocompatibility complex, class II, DR Alpha), *HLA-F* (major histocompatibility complex, class I, F), *CD44* (CD44 molecule), *IFI44* (interferon-induced protein 44) and *IFI44L* (interferon-induced protein 44 like) (Fig. 5b).

We identified 247 genes with sex-associated regulation in the long-term muscle meta-analysis (Fig. 6c). Their top functional ontology was chromatin organization (Supplementary Data 11) and many histone modifying genes were among the most differentially regulated with exercise and age. Other examples include the hypoxia-responsive transcription factor *HIF1A* (hypoxia inducible factor 1 alpha), and histone deacetylase 3 (*HDAC3*), which are important transcriptional regulators in skeletal muscle[51,52]. *MTMR3*, a lipid phosphatase that decreased more in male- vs. female-predominant cohorts, has to our knowledge not been previously associated with exercise. Although the decrease in *MTMR3* was small, it was consistent, which made us select this gene for validation, while observing that the meta-analysis result is in response to training and the validation cohort is in response to acute exercise. No significant interaction was observed in the validation cohort (Fig. 6d), emphasizing the need for the additional power provided by the meta-analysis to discover more subtle gene expression changes. The meta-analysis results can be easily queried through the online resource ExTraMeta for Exercise Transcriptome Meta-analysis at www. extrameta.org.

## Discussion
We took a meta-analysis approach to study the transcriptional landscape of acute and long-term exercise adaptation in human skeletal muscle and blood (ExTraMeta). Integration with interaction and regulation networks provided key insights into the molecular map of exercise responses. Gene Set Enrichment Analysis using the fold-changes inferred from a naive meta-analysis identified multiple relevant pathways with a marked overlap between the different analyses. Improving the inference at the gene level was done by fitting a model for each gene, accounting for potential moderators such as time, sex, training modality, and age. However, the gene-level overlap between the

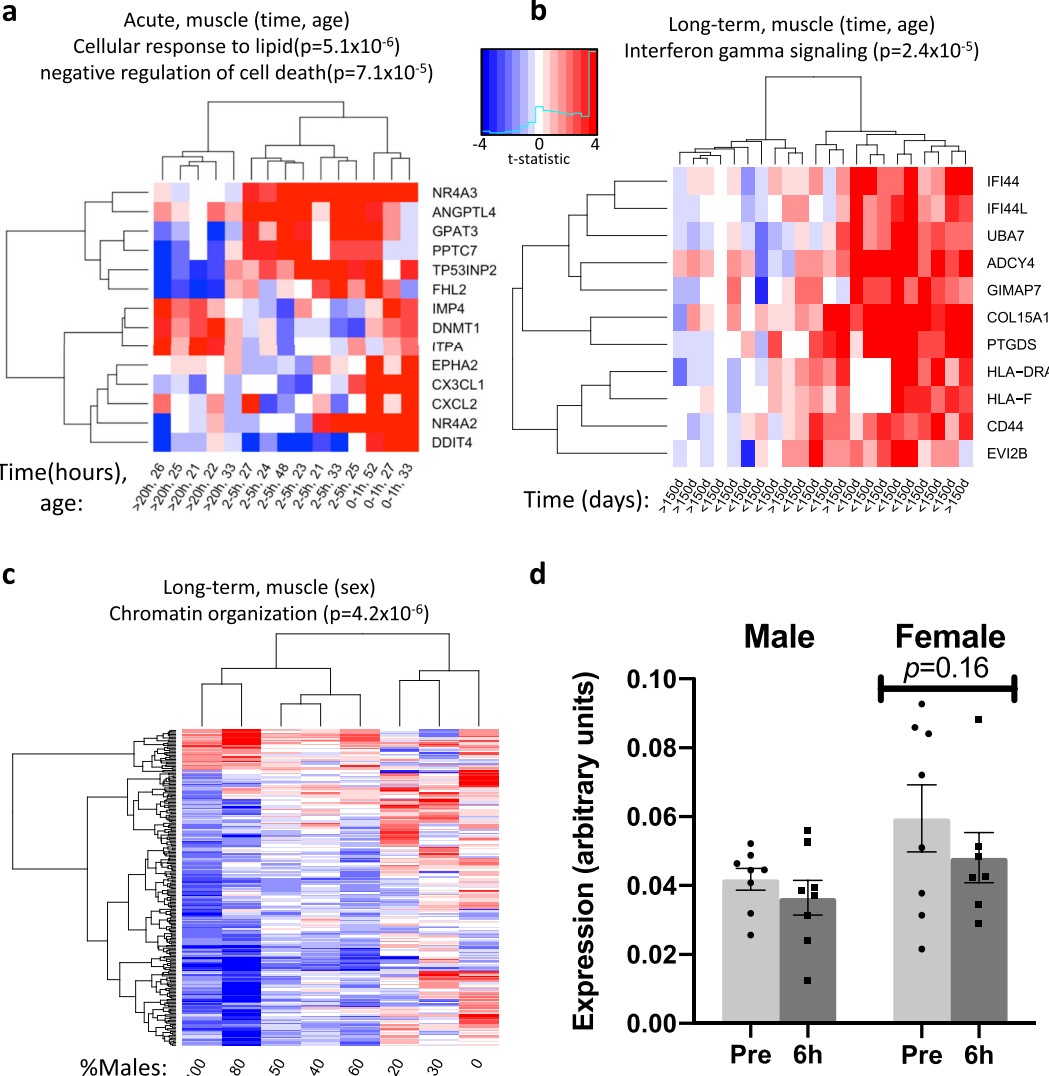

**Fig. 6 Covariate-specific transcriptional changes in response to exercise and training in human skeletal muscle. a** Heatmap of the 14 genes that were differentially regulated based on time and age after acute exercise. **b** Heatmap of the 11 genes that were differentially regulated based on time and age after long-term training. **c** Heatmap of the 247 genes that were associated with sex distribution within a cohort in response to long-term training. Each cell is the average t-statistic of a gene across a set of studies. **d** Skeletal muscle expression of *MTMR3* in males (*n* = 8) and females (*n* = 8) before and 6 h after an acute endurance exercise bout, analyzed with qRT-PCR in a separate validation cohort. Expression levels are calculated relative to the average of two housekeeping genes (*GAPDH* and *RPS18*). Data is presented as mean±standard error of the mean. Source data are provided as a Source Data file.

response to acute and chronic exercise was low, indicating that more data are required to further delineate the interactions by which the acute response transduces signals that translate into long-term effects.

For the muscular response to an acute exercise bout, we identified co-expressed gene clusters that provide insights into the differential temporal response patterns. The network analyses suggest information propagation through the underlying gene networks, with *SMAD3* as the main hub and a modality-independent central regulator. Integrating the time trajectories with known gene regulation networks (using DREM), predicted upstream regulators of the temporal response. The clustered gene sets were significantly enriched with pathways related to structural changes, angiogenesis, immune regulation and energy metabolism.

For long-term interventions in muscle, the response was dominated by structural changes and metabolic pathways. Our holistic approach identified a comprehensive response

interactome, but also highlighted that current curated pathways cover less than half of the identified genes, illustrating the existing knowledge gap and providing novel candidates for future research. Delving into moderator-associated responses, we identified sex- and age-associated differential expression, where long-term training induced a more prominent inflammatory response in older individuals. It is not clear how a greater inflammatory response at the transcriptional level affects the adaptation in older individuals, as inflammation is an essential part of exercise adaptation, and may also affect the degree of training-induced muscle damage.

Our meta-analysis substantially expands the catalog of sex-associated exercise responses, and offers multiple candidates for further research. We identified 247 genes that were associated with long-term training response and sex. They were functionally associated with histone modifications, which has been shown to be a key transcriptional regulation mechanism in response to endurance training in human skeletal muscle[53]. A summary of

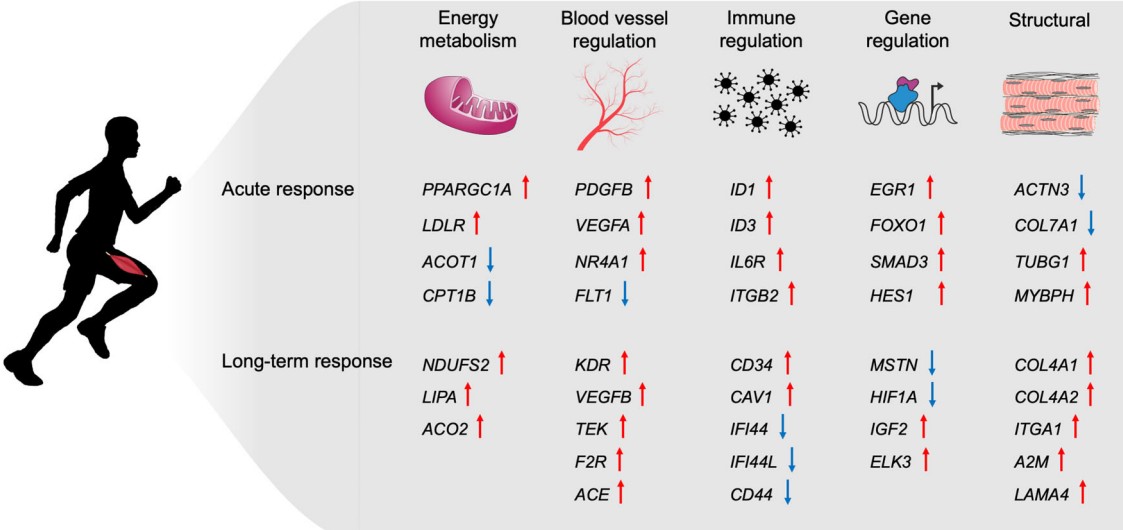

**Fig. 7 Overview of selected gene expression changes.** Summary of changes associated with key known adaptation mechanisms in skeletal muscle. Arrows indicate direction of change based on the base model from the meta-analysis. Source data are provided as a Source Data file.

selected skeletal muscle results is shown in Fig. 7. It is important to note that a limitation to these analyses was the inability to moderate for the physiological response in each individual, which is known to be heterogeneous[54].

In blood, the results were not as coherent as in skeletal muscle. We observed the following limitations of these data: (1) we were unable to perform a moderator-specific analysis due to lack of coverage of moderators, (2) less genes were included into the gene model selection process (e.g., only 8650 genes were considered in the acute response, see Methods for details), and (3) the fold-changes tended to be lower. Nevertheless, the base models identified 37 differential genes in the acute analysis, and 48 genes in the long-term analysis. These patterns were mainly enriched for immune responses, which is expected during recovery. However, the limited sample size and the inability to test for moderator effects due to low coverage (e.g., we could not find any public transcriptome datasets for acute resistance interventions) highlight where future research efforts are needed.

Recently, Pillon et al.[30] took a meta-analysis approach to analyze exercise and inactivity gene expression studies from skeletal muscle, and thousands of genes were identified to have differential expression. However, this study is limited in both power and false-positive control. First, the analysis accounted for only a single moderator and the dynamic response over time was not analyzed in a systematic way for all genes. Second, splitting the data likely resulted in loss of power for genes that respond to both resistance and endurance training. Finally, and most importantly, the analysis was based on a naive meta-analysis and selection of genes based on their p-values alone, and the effect heterogeneities were not inspected. When the number of studies is moderate or small (e.g., 15 or less, as in the exercise case) then p-values are often biased downwards[55,56]. This was also observed previously in the gene expression case, where selecting genes based on the p-value of the meta-analysis resulted in thousands of genes and in poor replicability even after false-discovery rate adjustment[28]. Meta-analysis methods have some additional limitations that should be considered, especially when studies are highly heterogeneous. First, random-effects meta-analysis does not directly explain variability. Second, meta-regression is often exploratory in nature and heavily relies on the included moderators. In our case we could not analyze potential moderators, including height, weight, BMI, training status, baseline VO₂max,

and baseline heart rate because the vast majority of the samples did not have these data available. Third, when the number of studies is moderate or small, small studies may be assigned too much weight[55,56]. Finally, meta-analysis can suffer from low power when the heterogeneity is high or when studies are sampled in an unbalanced way across their moderators[57,58].

We mitigated these issues by introducing a model selection pipeline that fits the moderator set for each gene to better explain the variability in the data. Moreover, we do not rely on p-values alone to select genes: we require that their models have both a high AICc difference and a high effect size (fold-change). As most moderator pairs are not correlated (Supplementary Fig. 3), we expect that the model selection process correctly detected the relevant moderators in most cases. While some specific models may not be accurate, our gene selection process was developed to highlight genes that are expected to manifest greater replicability. This is corroborated by the high concordance between the meta-analysis findings and the results from the validation cohort.

Understanding exercise response mechanisms holds immense potential for human health and medicine. Our analysis reveals differential expression trajectories in skeletal muscle together with their associated subnetworks and regulation events, and identifies sex- and age-associated transcriptional regulation. These results deepen our understanding of the transcriptional responses to exercise and provide a powerful, free, and easily accessible public resource (www.extrameta.org) for future research efforts in exercise physiology and medicine.

## Methods

**Data collection**. We inspected and annotated publicly available human exercise omics data available in the Gene Expression Omnibus[59] (GEO, search date 2/22/2019). The datasets were annotated by looking at their metadata taken from GEO or Recount[60], by extracting additional information from the publications (including supplementary data), and (whenever required) by personal communication with the authors. Only transcriptomics data had a reasonable number of studies to allow a meta-analysis of differential abundance. For these meta-analyses, we collected human blood and skeletal muscle gene expression datasets that had pre- and post-exercise samples for most of their subjects. Non-transcriptomics data, adipose tissue datasets, and datasets without multiple time points per subject were excluded from the analysis, even though some are available in the database (Supplementary Data 1). Each dataset was partitioned into cohorts based on tissue, study arm, and training modality. This was performed separately for acute bout datasets and long-term datasets (some studies had both). However, note that datasets GSE59088, GSE28998, GSE28392, and GSE106865 had both acute and

long-term cohorts. The included long-term studies ranged from six weeks to nine months of training.

To address missing sex information, we took a machine learning approach for imputation. We trained a linear support vector machine (SVM) classifier for sex using the expression levels from 469 Y chromosome genes that were covered by most of the platforms in our data. Studies from platforms GPL571, GPL17586, and GPL16686 were excluded due to low overlap with the other platforms. This affected a single cohort that was included in the meta-analysis: long-term cohort 34, see Supplementary Data 1. The training set contained 1382 samples with known sex information (851 males). Internal leave-study-out cross validation had very high performance with 0.99 area under the ROC curve and 0.96 area under the precision-recall curve. The predictions of final model (trained using all labeled data) were used for the meta-analysis.

**Preprocessing and moderators.** For each gene in each cohort we computed the mean and standard deviation of the log fold-change between the post-exercise time points and the baseline, illustrated in Fig. 1. We also kept the $p$-value of the pairwise test for each time point (i.e., vs. the pre-exercise samples). The result is a matrix of genes over the summary statistics (y—the mean log2 fold-change, v—the log2 fold-change variance, p—the paired $t$-test $p$-value). We excluded genes that had a missing value in 25% or more. This left >18,000 genes in each dataset (acute muscle—18,374, acute blood—18,621, long-term muscle—18,685, long-term blood —19,683).

For each cohort the following moderators (covariates) were collected: experiment type (long-term training program vs. single acute bout), tissue (blood vs. muscle), training modality (endurance, untrained, or resistance), sex measured as the proportion of males in the cohort, average age, and time (measured in hours for acute bouts and days for long-term training). The cohorts did not cover all combinations of the moderators. Moreover, the low-sample sizes, limited number of cohorts, and the observed heterogeneity, all posed challenges that may create inflation in false positives. Below we explain the different steps we took to select cohorts and moderators for the analysis where the goal was to both select relatively homogeneous cohorts and allow exploration of some moderators that were reasonably covered.

Exercise vs. untrained cohorts: we first observed that most exercise studies did not monitor the gene expression change in untrained controls. We kept these cohorts separately (3 for acute, muscle; 2 for acute, blood, and 2 for long-term, muscle, <30 subjects in each case) and used them to validate the selected genes from the meta-analyses (Fig. 3c).

Next, for the exercise cohorts we examined the moderators in each case. The number of blood cohorts and their coverage was poor: acute blood data had seven cohorts and they were all from endurance training experiments, whereas long-term blood had three cohorts only. For the muscle data we decided to include the following moderators in each analysis:

1. Acute, muscle (15 cohorts): time, training, and age. Time was partitioned into windows as in the acute blood analysis above. Sex was excluded as only four cohorts had females.
2. Long-term, muscle (26 cohorts): all moderators. Time was binned into <150 days, and >150 days.

**Random-effects meta-analysis.** We used the R metafor package to perform meta-analysis and meta-regression for each gene[61]. In both cases, we model the fold-change of a gene. Assume we are given $y_i$, $i \in 1,..., k$ estimated effects of a gene, where $i$ denotes a specific post-exercise time point in a cohort (nested within studies). Let $y_i$ be the log fold-change with variance $v_i$. For the standard random-effects model we assume that $y_i = \mu + u_i + \epsilon_i$, where $y_i$ is the observed effect, $u_i$ is the true (unknown) effect $u_i \sim N(0, \tau^2)$, and $\epsilon_i$ is the sampling error term, $\epsilon_i \sim N(0, v_i)$, which are assumed to be independent. In this analysis we were interested in both the fold-change of the gene across all cohorts and time points ($\mu$), and the total amount of true heterogeneity among the true effects ($\tau^2$). For an interpretable statistic of consistency we used $I^2$: the ratio of true heterogeneity to total observed variation[62].

For random effects we also tested a nested model structure, denoted as *cohort | study* (i.e., *inner | outer*). Thus, here the random effects have a block-wise structure: observations from different studies will be independent, but observations from cohorts from the same studies will be dependent.

Random-effects meta-analysis was used both as a base model to evaluate the effect and heterogeneity in the exercise meta-analyses (1–4 above), and to evaluate the fold-change of a gene in the untrained controls. The goal of analyzing the untrained controls was to evaluate genes that are differential but not because of the exercise (e.g., due to the process of taking a muscle biopsy).

**Meta-regression.** We used mixed-effects meta-regression to extend the analysis to include a set of effect moderators (i.e., covariates). In this case, assuming that there is a large unexplained heterogeneity among the true effects, we model the fold-change of a gene using:

$$y_i = \beta_0 + \sum_{j=1}^{p} \beta_j x_{i,j} + u_i + \epsilon_i \qquad (1)$$

where $x_{i,j}$ denotes the value of the j-th moderator variable for the i-th effect, $u_i$ represents a random effect, whose variance represents the residual heterogeneity among the true effects. In other words, it is the variability among the true effects not captured by the moderators in the model. The different tested distribution structure of $u_i$ are the same as in the meta-analysis above.

**Model selection.** Whenever meta-regression was considered, we tested all possible combinations of a subset of the moderators (including no moderators and all moderators as options). This was combined with testing two different random-effects structures as discussed above. For each model we computed the AICc score, which is more suitable for our case with a limited number of studies as compared to other alternatives[35,36]. We then ranked all models by their AICc and excluded models that were not among the top two or were not base models (i.e., models without moderators).

**Gene selection process.** To reduce the number of model selection analyses and focus on genes that are likely to be replicable, we first excluded genes that had no or only a single $p$-value < 0.05 across all cohorts. For the remaining genes (acute, muscle: 13,016; long-term,muscle: 14,309; acute,blood: 8650; long-term,blood: 508) we computed the base models and performed the model selection analysis. A model was considered a significant improvement over a base model (and thus selected) only if $\Delta AICc > 5$, $p < 0.001$, $\exists \beta_j; |\beta_j| > \mu_0$, where $p$ is the $p$-value of the model (including all moderators), and $\mu_0$ is a threshold for absolute effect size. We used $\mu_0 = 0.1$. In case that no improvement over the base model was identified, the base model was selected only if it had $I^2 < 50\%$, $p < 0.001$, $|\mu| > \mu_0$, where $p$ is the $p$-value of the estimated effect and $\mu_0$ is defined as above. Note that using both thresholds for the model significance and the effect size is recommended as it tends to reduce false positives and highlight reproducible results[28].

**Gene clustering.** In each of the four analyses (e.g., acute muscle) we took all models that had the same covariates and clustered their t-statistic matrices using $k$-means[63]. In each such analysis we determined the number of clusters using the elbow method: we plot the within-cluster sum of squares as a function of the number of clusters (testing 1–15) taking the last point that had at least 60% improvement from the previous one as the number of clusters.

For the time course model detected in acute muscle we also used DREM for clustering and network inference for transcription factors (TFs)[45]. We used the mean log fold-change per gene in each time point as input. The analysis was performed on the acute muscle time-associated genes. We used DREM with the ENCODE TF-target network[64] and a low penalty for adding nodes. We lowered the default of 40 until the detected cluster trajectories fitted our $k$-means clustering results (with 12 as the selected value).

**Functional analysis and network visualization.** GO enrichment analysis was performed using topGO[65]. Reactome enrichment analysis was performed using ReactomePA[66]. Network analysis for selected gene sets was completed using GeneMANIA[67] in Cytoscape[68,69]. Visualization of Reactome networks in Cytoscape was done using Reaction FI[70]. GSEA was run using the fGSEA[71] R package using Reactome pathways with at least 20 genes and up to 200 genes. We set the number of GSEA permutations to 50,000.

**Validation cohort and qRT-PCR.** A separate acute endurance exercise cohort of 16 individuals, 8 males and 8 females, was included to validate some of the meta-analysis findings. The study was approved by the Regional Ethical Review Board in Stockholm, Sweden, and was conducted in accordance with the Declaration of Helsinki. All individuals provided written informed consent. The detailed study design has been described elsewhere[72]. In brief, the subjects performed 60 min of endurance exercise on a cycle ergometer at a workload corresponding to 70% of their peak VO₂. Skeletal muscle biopsies from *vastus lateralis* were obtained before exercise, and at 2 h and 6 h after the exercise bout. Skeletal muscle samples were homogenized using a bead homogenizer and total RNA was extracted by the Trizol® method (Invitrogen, Carlsbad, CA, USA), according to the manufacturer's specifications. One ug RNA was reverse transcribed using Superscript reverse transcriptase (Life Technologies) and random hexamer primers (Roche Diagnostics). Quantitative real-time PCR was performed with primers for *SMAD3*, *NR4A1*, *SCN2B*, *HES1*, *ID1*, *SLC25A25*, *MTMR3* and normalized to the average of two housekeeping genes that were stable across the intervention: glyceraldehyde-3-phosphate dehydrogenase (*GAPDH*) and ribosomal protein S18 (*RPS18*) (Sigma-Aldrich, all primer sequences are listed in Supplementary Table 1). Expression level was calculated using the $\Delta C_T$ method and statistical analysis performed using a mixed-effects model followed by Dunnett's multiple comparisons test.

**Reporting summary**. Further information on research design is available in the Nature Research Reporting Summary linked to this article.

## Data availability

The curated database (ExTraMeta) generated in this study is available at www.extrameta.org. Data supporting the findings of this study are available within the article and its supplementary information files. Source data for the meta-analysis are provided with this paper. All computed summary statistics are available here: https://github.com/AshleyLab/motrpac_public_data_analysis Source data are provided with this paper.

## Code availability

The code for all analyses presented in this paper is available here: https://github.com/AshleyLab/motrpac_public_data_analysis

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

## Acknowledgements
This research was supported by the NIH Common Fund (award number U24OD026629) and the Knut and Alice Wallenberg Foundation (M.E.L.).

## Author contributions
Conception and design: D.A., M.E.L., M.T.W., M.R., E.A.A. Analysis and interpretation: D.A., M.E.L., M.T.W., E.A.A. Validation experiments: J.N. Manuscript draft: D.A., M.E.L. All authors contributed to the research of the published work, and have read and approved the final manuscript.

## Competing interests
The authors declare no competing interests.
