## [Peer Review File · Nature Communications]

Reviewers' Comments:

Reviewer #1:

Remarks to the Author:

In „Time trajectories in the transcriptomic response to exercise“ Amar et al. present computational framework comprising data from 43 studies with 739 individuals with gene expression data from skeletal muscle and plasma from before, immediately after and after chronic exercise training. Using linear mixed effects meta regression models new time patterns and regulatory networks were identified. Specifically, SMAD3 was shown to be an exercise modality “regulator” of the exercise response. The authors claim that these “results deepen our understanding of the transcriptional responses to exercise and provide a powerful resource for future research efforts in exercise physiology and medicine”. The database used in this submission provides enormous potential and the authors are to be applauded for their efforts in collecting and combining data from 43 studies. Unfortunately, however, the manuscript is poorly written and the figures are not really helpful in communicating the main message. I have outlined some specific comments below.

Major comments:

- in the abstract you claim that physical activity is the most important determinant of physical performance. I guess with “physical performance” you mean cardiorespiratory fitness. If so, than you ought to acknowledge that physical activity and cardiorespiratory fitness are two completely different topics
- Your logic in the introduction is unclear to me. For example your sentence “Although resistance and endurance training lead to distinct cellular adaptation, there are several genes, including known regulatory factors, that are differentially expressed in response to both modalities, for example VEGFA , PDK4 , PPARGC1A and FOXO3.” What is point you are trying to make? MCT and resistance training lead to different phenotypes through the same pathways, but there are also differences?!
- You included four confounders -> Could you please provide a justification (e.g. using directed acyclic graphs) as to why you decided that these confounders are appropriate? For example, why did you decide to not include training status?
- you also filtered the GO database for adipose tissue but this is not reported in the manuscript.
- Figure 2B is not helpful. This needs to be better explained. What do the abbreviations for the different cohorts mean? Where is the phenotypic description of these cohorts? What MRPL34 the only significant findings? You report a high heterogeneity for this gene. Yet, this may simply be an effect of time and not heterogeneity between studies per se.
- Table 1 is not helpful. The gene ranks are hard to read and to interpret.
- You use the term “clinical heterogeneity” – what do you mean?
- the abbreviations in Figure 3A and 3B are also not intuitive and should have explained in the captions

Minor comments:

- please spell out gene names at first use
- please always spell out abbreviations at first use and add the abbreviation in parenthesis thereafter
- please provide IDs (e.g. KEGG) for the significant pathways

References

1. DeFina LF, Haskell WL, Willis BL, et al. Physical activity versus cardiorespiratory fitness: two (partly) distinct components of cardiovascular health? Progress in cardiovascular diseases 2015; 57: 324-329. DOI: 10.1016/j.pcad.2014.09.008.

Reviewer #2:

Remarks to the Author:

In the paper, the authors conducted a meta-analysis to look for gene expression change before and after acute exercise or long-term training from skeletal muscle (n=592) or blood (n=199).

- It is a meta-analysis paper. I am confused what are major findings, if any. Also, there are some inconsistencies in the paper which are detailed below. And many figures presented are not very informative.
- In GSEA analysis, what pathway/gene set categories were tested?
- Is NES normalized enriched score? Pathways in Table 1 should be sorted by NES, right?
- MTMR3 is mentioned in the abstract as a novel exercise-associated gene. But no supporting data is presented except that it is one of the 247 genes identified in a meta-analysis. Why select this one out of the 247? What is its rank among the 247? What is the p-value? "Although the decrease in MTMR3 was small, it was consistent and novel," What do you mean a decrease is "novel"?
- The authors mentioned validation study using qRT-PCR on seven genes (page 26). Figure 4A shows only six. MTMR3 is missing. And I am not very clear how many of these are validated? MTMR3 is not validated, correct? So this gene should not be claimed as a novel finding as claimed in the Abstract.
- GO enrichment analysis is mentioned in Methods, but never mentioned in Results. GSEA is mentioned in Results, but not described in Methods.
- Figure 3D. what are the columns of the heatmap?
- Figure 6B. Some of the columns do not have labels. 6C. what are the numbers at the bottom mean?

REVIEWER COMMENTS

Reviewer #1 (Remarks to the Author):

In „Time trajectories in the transcriptomic response to exercise” Amar et al. present computational framework comprising data from 43 studies with 739 individuals with gene expression data from skeletal muscle and plasma from before, immediately after and after chronic exercise training. Using linear mixed effects meta regression models new time patterns and regulatory networks were identified. Specifically, SMAD3 was shown to be an exercise modality “regulator” of the exercise response. The authors claim that these “results deepen our understanding of the transcriptional responses to exercise and provide a powerful resource for future research efforts in exercise physiology and medicine”. The database used in this submission provides enormous potential and the authors are to be applauded for their efforts in collecting and combining data from 43 studies. Unfortunately, however, the manuscript is poorly written and the figures are not really helpful in communicating the main message. I have outlined some specific comments below.

Major comments:

- in the abstract you claim that physical activity is the most important determinant of physical performance. I guess with “physical performance” you mean cardiorespiratory fitness. If so, than you ought to acknowledge that physical activity and cardiorespiratory fitness are two completely different topics¹

We thank the reviewer for pointing out this important distinction. In the abstract, we meant physical performance in a very broad sense, including both endurance performance (mainly determined by cardiorespiratory fitness, as well as e.g. running economy if it is a running-based performance) and strength performance, both of which have been important attributes for human survival. To make the distinction between physical activity and cardiorespiratory fitness clearer, we have added a sentence to the introduction, as well as a reference to the excellent review article suggested by the reviewer. In order to shorten the abstract and avoid the concept of performance, which is not the main purpose of this study, we have removed that first part of the initial sentence.

- Your logic in the introduction is unclear to me. For example your sentence “Although resistance and endurance training lead to distinct cellular adaptation, there are several genes, including known regulatory factors, that are differentially expressed in response to both modalities, for example VEGFA , PDK4 , PPARGC1A and FOXO3.” What is point you are trying to make? MCT and resistance training lead to different phenotypes through the same pathways, but there are also differences?!

We agree that this sentence is unclear and confuses the main message. We thank the reviewer for pointing this out and have removed this part of the introduction.

We have also rearranged the introduction to first discuss the biological background and motivation (paragraphs 1-2, lines 77-101 in track changes version of manuscript) and then the statistical/computational background (paragraph 3, lines 102-140 in track changes

version of manuscript). This was done to clarify the introduction and study rationale for the reader.

- You included four confounders -> Could you please provide a justification (e.g. using directed acyclic graphs) as to why you decided that these confounders are appropriate? For example, why did you decide to not include training status?

We thank the reviewer for raising this important point. Indeed our previous version did not put enough emphasis on explaining it. While we manually parsed many important covariates from the datasets, for most of them (including height, weight, BMI, training status, baseline VO2 max, baseline heart rate, etc.), sufficient data were not available from these publications to be included in our meta-regression models. We explain this point in the Results, lines 189-191. We further elaborate, with emphasis on the limitations of these missing data, in the Discussion, lines 700-704 in track changes version of manuscript.

- you also filtered the GO database for adipose tissue but this is not reported in the manuscript.

This point is explained in Materials and Methods, Data collection (lines 742-743 in track changes version of manuscript) and reads:

“Non-transcriptomics data, adipose tissue datasets, and datasets without multiple time points per subject were excluded from the analysis, even though some are available in the database (Supplementary Data 1).”

- Figure 2B is not helpful. This needs to be better explained. What do the abbreviations for the different cohorts mean? Where is the phenotypic description of these cohorts? What MRPL34 the only significant findings? You report a high heterogeneity for this gene. Yet, this may simply be an effect of time and not heterogeneity between studies per se.

We thank the reviewer for pointing out the issues around this figure. The goal of this figure is to motivate the model selection analysis presented later in the paper. Indeed, as can be seen in the Figure, and as correctly pointed out in the comment above, the negative fold changes are associated with time, which can be missed in a meta-analysis that does not utilize time as an additional moderator.

We address this comment both by improving the figure and by changing the text. The forest plots in the paper (including Figure 2B) now show the cohort metadata together with the fold change confidence interval. These data include the cohort id, number of samples (N), exercise type (EE - endurance exercise, RE - resistance exercise), mean age, proportion of males, and time window.

The figure legend now explicitly explain the goal of the figure and the abbreviations:

“B) Forest plot of the effect sizes of MRPL34 in acute muscle cohorts illustrates the importance of adding time as a moderator. Rows represent the fold change of a cohort in a given time point. Down-regulation is observed only in time points >20 hours. Cohort: the ID given to the cohort in this study (see Supplementary Data 1 for details), N: sample size, Type: exercise type, RE: resistance exercise, EE: endurance exercise, Age: mean age in cohort, %M: percent of males in cohort, Time:

the time window in hours. Note that a cohort can have multiple rows with the same time window and different fold changes (e.g., if a study measured both 2h and 4h then both time points will be assigned 2-5h).”

- Table 1 is not helpful. The gene ranks are hard to read and to interpret.

We agree with the reviewer and have therefore decided to make Table 1 into a supplementary figure. This both avoids the confusion from over-emphasizing these results in the main text and allows us to increase the table size.

- You use the term “clinical heterogeneity” – what do you mean?

We use the term clinical heterogeneity to describe the differences between cohorts with regards to both phenotypic and intervention heterogeneity. The main differences are detailed in Supplementary Data 1, for example average subject age, sex distribution and post-exercise time points in a study. However, a lot of information was not available, including baseline fitness levels, BMI etc. We appreciate this comment and have clarified what we mean by clinical heterogeneity in the results section, line 301-302 in track changes version of manuscript.

- the abbreviations in Figure 3A and 3B are also not intuitive and should have explained in the captions

As mentioned above, we have modified all forest plots to include additional phenotypic data and updated the figures accordingly. We also explain all abbreviations in the figure legends.

Minor comments:

We thank the reviewer for these comments to improve the manuscript and have addressed them all throughout text.

- please spell out gene names at first use

We have addressed this throughout the manuscript.

- please always spell out abbreviations at first use and add the abbreviation in parenthesis thereafter

We have addressed this throughout the manuscript.

- please provide IDs (e.g. KEGG) for the significant pathways

For GSEA we used the fgsea R package that uses Reactome pathways. This package does not output the Reactome ids, but alternatively provides the Reactome full names, which are unique identifiers. We now provide all GSEA results in Supplementary Data 4.

References

1. DeFina LF, Haskell WL, Willis BL, et al. Physical activity versus cardiorespiratory fitness: two (partly) distinct components of cardiovascular health? *Progress in cardiovascular diseases* 2015; 57: 324-329. DOI: 10.1016/j.pcad.2014.09.008.

Reviewer #2 (Remarks to the Author):

In the paper, the authors conducted a meta-analysis to look for gene expression change before and after acute exercise or long-term training from skeletal muscle (n=592) or blood (n=199).

- It is a meta-analysis paper. I am confused what are major findings, if any. Also, there are some inconsistencies in the paper which are detailed below. And many figures presented are not very informative.

We thank the reviewer for bringing this important point to our attention. We have re-written parts of the manuscript to highlight the major findings, moved minor results and their display items to the supplementary material, and edited several figures to make them more informative. We have emphasized the major findings in the updated abstract and text to clarify how we provide a powerful resource that expands the transcriptional landscape of exercise adaptation by extending previously known responses and their regulatory networks, and identifying novel modality-, time-, age-, and sex-associated changes. In addition, we have addressed all the points detailed in the comments below.

- In GSEA analysis, what pathway/gene set categories were tested?

For GSEA we used the `fgsea` R package that uses Reactome pathways. We now provide all GSEA results in Supplementary Data 4, and we also explain how GSEA was run in the Methods (see lines 890-891 in track changes version of manuscript).

- Is NES normalized enriched score? Pathways in Table 1 should be sorted by NES, right?

NES are indeed the normalized enrichment scores. Table 1, which is now Supplementary figure 1, shows the top 3 up- and down- regulated pathways sorted by their NES.

- *MTMR3* is mentioned in the abstract as a novel exercise-associated gene. But no supporting data is presented except that it is one of the 247 genes identified in a meta-analysis. Why select this one out of the 247? What is its rank among the 247? What is the p-value? "Although the decrease in *MTMR3* was small, it was consistent and novel," What do you mean a decrease is "novel"?

Very few exercise studies have had the power to identify sex-associated gene expression changes with training, so from that perspective most of the 247 genes are novel. However, many of them have been reported to change with training in studies including one sex or from combined analyses of both sexes. The reason for highlighting *MTMR3* in the manuscript and describing it as novel was that it is a lipid-phosphatase that has not been

reported to change with training in previous literature. We did include validation of *MTMR3*, and a graph showing that data is now included in Figure 6 (please see specific response below). In accordance with the reviewer's suggestion, we have also removed the specific mention of *MTMR3* in the abstract.

- The authors mentioned validation study using qRT-PCR on seven genes (page 26). Figure 4A shows only six. *MTMR3* is missing. And I am not very clear how many of these are validated? *MTMR3* is not validated, correct? So this gene should not be claimed as a novel finding as claimed in the Abstract.

We thank the reviewer for this important observation. We did validate *MTMR3*, although we only included the data in written format and not as a figure. It was not included in Figure 4 because it was identified as a sex-specific gene with long-term training, as opposed to the acute exercise changes described in Figure 4. Although the acute exercise intervention applied in the validation cohort was not ideal to investigate changes with long-term training, where *MTMR3* was significant, we still wanted to include it. The data comparing pre exercise with the 6 hour time point (closest to the 24 and 48 hours common for the training intervention cohorts of the meta-analysis) for men and women has been added to Figure 6. Despite the low number of subjects (7 women and 8 men), it shows a borderline significant decrease specifically in women, which was observed in the meta-analysis. This demonstrates how the added power of the meta-analysis is key to discover small, but significant changes that could have important biological implications, for example in understanding the different adaptation mechanisms that are important in men versus women.

- GO enrichment analysis is mentioned in Methods, but never mentioned in Results. GSEA is mentioned in Results, but not described in Methods.

GO enrichment analysis of our selected gene sets is discussed in the Results lines 353-355. We now added the description of our GSEA pipeline in the Methods, see lines 890-891 (lines refer to the track changes version of manuscript).

- Figure 3D. what are the columns of the heatmap?

We added an explanation in the figure legend, which now reads:

“d Heatmap of the up-regulated genes selected in the acute blood meta-analysis. Columns represent cohorts and rows are genes. All acute blood studies measured the response to endurance exercise in up to 3 hours after the bout ended.”

- Figure 6B. Some of the columns do not have labels. 6C. what are the numbers at the bottom mean?

We added labels to all columns in this figure with both a label in the figure and an explanation in the figure legend. For example, Figure 6C now explicitly shows that the numbers are the percent of males in the studies:

a Acute, muscle (time, age)
 Cellular response to lipid ($p=5.1 \times 10^{-6}$)
 negative regulation of cell death ($p=7.1 \times 10^{-5}$)

b Long-term, muscle (time, age)
 Interferon gamma signaling ($p=2.4 \times 10^{-5}$)

c Long-term, muscle (sex)
 Chromatin organization ($p=4.2 \times 10^{-6}$)

d

Reviewers' Comments:

Reviewer #1:

Remarks to the Author:

With their revision of „Time trajectories in the transcriptomic response to exercise“ Amar et al. provide a significantly improved manuscript. Albeit, I have to say that I am rather surprised about this revision after both reviewers provided such critical feedback. With this being said, the current version is much better but still has some significant shortcomings which I have outlined below.

MAJOR COMMENTS:

1. The authors mix very different concepts. Physical (in)activity and exercise training are very different things. If the authors set out to explore the effects of physical inactivity (e.g. in the first sentence of their abstract), muscle sample from bed rest studies would have been appropriate. Yet, here the authors provide a resource the effects of acute and chronic exercise (most likely mainly in relatively young study participants). This needs to be corrected throughout the manuscript.
2. I would encourage the authors to read some of the recently published 2-sample MR studies and the Generation 100 study (<https://www.bmj.com/content/371/bmj.m3485>). While I personally also “believe” that physical activity is a good thing, the currently available data does not always support this.
3. You found a large amount of heterogeneity for some of the identified/highlighted genes. Why do you believe that these are still valid findings which have causal inference? Would not some sort of cut-off for I2 be appropriate?
4. I am surprised by the very low number of genes associated with acute response to exercise. Out of the 18,000 genes only 38 genes were differently expressed in the blood. How do you explain such a low number – simply considering the massive surge in immune cells due to physical exertion I would have expected a number closer to what was found in the skeletal muscle (537 genes).
5. You spend a lot of time discussing PGC1a. PGC1a is simply an energy regulator (<https://www.ncbi.nlm.nih.gov/pmc/articles/PMC3057551/>) and while your findings validate your results, it is certainly not novel.
6. In the first paragraph of the discussion you state that the low level of overlap between acute and chronic exercise challenges deserves further research – those are two completely different settings?! They should be different as one represents response to an acute stress and the other one chronic adaptation.
7. I also wonder why you separated muscle and blood. I believe it would be very interesting to assess the changes in one sample compared to changes in the other. Would it be possible to run e.g. a pearson correlation between changes in GE in skeletal muscle compared to blood?
8. You mention the study by Pillon and heavily criticize their approach. While you may be correct in the shortcomings of their methods, I wonder why you never compared your results with theirs. Just out of curiosity it may be interesting.
9. A very large shortcoming of the presented manuscript is that the GEO was searched more than 18 months ago. A current search must be performed if the authors aim to publish their findings in such a high impact journal as “Nature Communications”.
10. In you methods section you spend a lot of time explaining meta-regression to the reader in textbook fashion. This section should be shortened. You also highlight how your AIC approach is more suitable for small sample size. While this is correct, you also only have a relative comparison between models. Could you please provide a R2 for the identified genes?
11. Another important drawback of your qPCR analysis in the validation cohort is that you used GAPDH as a control. GAPDH should not be used as a housekeeping gene when performing exercise studies. Exercise training influence the glycolysis of immune cells. Please consider these two manuscripts when identifying appropriate controls:

-

https://journals.physiology.org/doi/full/10.1152/physiolgenomics.00025.2005?utm_source=TrendMD&utm_medium=cpc&utm_campaign=Physiological_Genomics_TrendMD_0

- <https://pubmed.ncbi.nlm.nih.gov/12184808/>

Minor comments:

12. Why do you capitalize HDL and LDL in the introduction?
13. Why do you italic "moderators" in the introduction?
14. you always present the acronym of the genes first followed by the spelled-out name. Please check with the journal formatting guidelines whether this is appropriate
15. please spell out BMI at first use
16. please spell out VO2max at first use
17. p16 ln 446 it is "cohorts".

Reviewer #2:

Remarks to the Author:

The authors have addressed all my comments. I am okay with most of the answers. For Figure 3D, I still suggest adding labels for columns in the heatmap.

RESPONSE TO REVIEWERS

Reviewer #1 (Remarks to the Author):

With their revision of „Time trajectories in the transcriptomic response to exercise” Amar et al. provide a significantly improved manuscript. Albeit, I have to say that I am rather surprised about this revision after both reviewers provided such critical feedback. With this being said, the current version is much better but still has some significant shortcomings which I have outlined below.

MAJOR COMMENTS:

1. The authors mix very different concepts. Physical (in)activity and exercise training are very different things. If the authors set out to explore the effects of physical inactivity (e.g. in the first sentence of their abstract), muscle sample from bed rest studies would have been appropriate. Yet, here the authors provide a resource the effects of acute and chronic exercise (most likely mainly in relatively young study participants). This needs to be corrected throughout the manuscript.

We agree with the reviewer that this is an important distinction. To avoid confusion by mentioning lack of physical activity, which is of course not studied here, we have removed that and instead only refer to exercise or exercise training. This has been addressed throughout the manuscript.

2. I would encourage the authors to read some of the recently published 2-sample MR studies and the Generation 100 study (<https://www.bmj.com/content/371/bmj.m3485>). While I personally also “believe” that physical activity is a good thing, the currently available data does not always support this.

We thank the reviewer for highlighting these studies, and by no means do we want to convey the message that exercise is beneficial for all disease conditions. We have added references to some of the mendelian randomization studies in the introduction. While most of those studies do establish the causal effect of exercise in the reduction of risk for depression, breast and colorectal cancer for example, they show no effect on prevention of Parkinson's disease or schizophrenia, which we now mention. Further, MR-based causal estimates that are either with an opposite sign or insignificant do not necessarily contradict the proposition that physical activity is beneficial. Most MR studies are limited in that they typically estimate a single causal effect for a given population and they may also rely on statistically weak genetic instruments. For example, if effects are moderated by age (e.g., a non linear scenario in which a treatment may be useful for young adults but harmful for older populations) then the MR analysis on the entire population may be inaccurate and even biased as this type of interaction is not considered in the standard MR model.

The Generation 100 study mentioned by the reviewer is an immense effort to investigate potential benefits of high intensity training compared to moderate intensity training. However, it was published after our resubmission date and was therefore not included. Also, the control group in this study was not untrained as they were encouraged to follow recommended guidelines, and actually ended up training at an intensity higher than the moderate intensity group. Hence it is very difficult to draw conclusions about the potential effects of training on premature mortality based on this study as all groups were trained.

3. You found a large amount of heterogeneity for some of the identified/highlighted genes. Why do you believe that these are still valid findings which have causal inference? Would not some sort of cut-off for I2 be appropriate?

As we explain in the Methods, subsection “Gene selection process”, we use the I2 score as a part of our model selection process per gene. Specifically, base models with I2>50% were considered unreliable and were not used. For non-base models we required that they will manifest a marked improvement in their AIC score. This is a conservative approach that aims to focus on genes with extreme improvement in their explained variance when adding moderators to the model. In all cases, genes were selected only if they passed another threshold on the estimated fold changes.

The set of filters above was designed to reduce the number of false positives and improve the ability to identify causally relevant genes. However, we discuss the limitations of both the study and the currently available exercise transcriptomic data at length throughout two paragraphs (Discussion paragraphs 6-7).

4. I am surprised by the very low number of genes associated with acute response to exercise. Out of the 18,000 genes only 38 genes were differently expressed in the blood. How do you explain such a low number – simply considering the massive surge in immune cells due to physical exertion I would have expected a number closer to what was found in the skeletal muscle (537 genes).

We thank the reviewer for raising this important issue and highlighting that we did not explain this issue clearly enough in the paper. We have addressed it as explained below.

Briefly, there are several reasons for the relatively low number of discoveries in blood. First, note that as we explain in the Methods, subsection “Gene selection process”, we excluded genes that had no or only a single p-value < 0.05 across all cohorts. This resulted in the following numbers per analysis: acute,muscle: 13,016; long-term,muscle: 14,309; acute,blood: 8650; long-term,blood: 508. These numbers suggest that the blood datasets tend to have lower statistical power compared to the muscle studies. Second, we observe that the blood studies tend to have lower fold changes. For example, when examining the absolute fold change of all base models, only 470 genes passed our 0.1 threshold in acute,blood. In contrast, 2,135 genes passed this filter in the acute,muscle datasets. In this study, we decided to put emphasis on genes that are likely to be replicable using the guidelines suggested in previous gene expression studies (see Sweeney et al. 2017 and Amar et al. 2017). These studies proposed such conservative guidelines to avoid unreasonable false positive rates.

To explain these points we added the following text:

- *Results:*

“We detected differential genes in each of the four analyses: acute exercise, muscle: 537 genes; long-term training, muscle: 441 genes; acute exercise, blood: 37 genes; and long-term training, blood: 48 genes. The low number of genes from the blood analyses is due to several factors: (1) lower number of cohorts, (2) limited statistical power (e.g., only 8650 genes had at least two p-values <0.05 in the acute datasets as compared to 13,016 in muscle), and (3) lower effect sizes (e.g., only 470 genes had fold change >0.1 in the acute,blood base models).”

- *Methods, subsection “Gene selection process”:*

“To reduce the number of model selection analyses and focus on genes that are likely to be replicable, we first excluded genes that had no or only a single p-value < 0.05 across all cohorts. For the remaining genes (acute,muscle: 13,016; long-term,muscle: 14,309; acute,blood: 8650; long-term,blood: 508) we computed the base models and performed the model selection analysis. “

- *Discussion:*

“In blood, the results were not as coherent as in skeletal muscle. We observed the following limitations of these data: (1) While we were unable to perform a moderator-specific analysis due to lack of coverage of moderators, (2) less genes were included into the gene model selection process (e.g., only 8,650 genes were considered in the acute response, see Methods for details), and (3) the fold changes tended to be lower. Nevertheless, the base models identified 37 differential genes in the acute analysis, and 48 genes in the long-term analysis. “

5. You spend a lot of time discussing PGC1a. PGC1a is simply an energy regulator (<https://www.ncbi.nlm.nih.gov/pmc/articles/PMC3057551/>) and while your findings validate your results, it is certainly not novel.

PGC1a is one of the most well-studied genes in the exercise literature and is given as an initial example for the reader, illustrating that the analysis presented models this well-known exercise gene correctly. We do not claim anywhere in the paper that this is a novel discovery. On the contrary, we explicitly write:

“To illustrate the output of the algorithm, consider the response of *PPARGC1A* (PPARG Coactivator 1 Alpha or PGC-1 α) in skeletal muscle after acute exercise. **This gene is an exercise-mediated regulator of muscle adaptation with a well-documented transcriptomic response**³³” Hence, we definitely agree with the reviewer and by no means do we claim that PGC1a is novel, only confirmatory.

6. In the first paragraph of the discussion you state that the low level of overlap between acute and chronic exercise challenges deserves further research – those are two completely different settings?! They should be different as one represents response to an acute stress and the other one chronic adaptation.

We agree with the reviewer that acute and chronic exercise represent different settings, and very different time resolution. The statement that is referred to in the discussion (*However, the gene-level overlap between the response to acute and chronic exercise was low, indicating that more data are required to further delineate the interactions by which the acute response transduces signals that translate into long-term effects.*) is simply meant to highlight that the interaction between the acute and long-term responses need further research, for example by increased time resolution of post-exercise sampling and studies that look at the response of a few days of exercise, where the current studies are mostly either analyzing the acute response or the response to weeks or months of training. Also, several studies highlight that it is the repeated acute exercise bouts that lead to long-term adaptation also at the transcriptional level (see for example Perry et al. 2010. “Repeated Transient mRNA Bursts Precede Increases in Transcriptional and Mitochondrial Proteins during Training in Human Skeletal Muscle.” *The Journal of Physiology* 588 (Pt 23): 4795–4810.).

7. I also wonder why you separated muscle and blood. I believe it would be very interesting to assess the changes in one sample compared to changes in the other. Would it be possible to run e.g. a pearson correlation between changes in GE in skeletal muscle compared to blood?

We thank the reviewer for raising this question and correctly pointing out that comparing the tissues is interesting.

We separated the analyses into the four groups for two main reasons: first, merging the two tissues will exacerbate the high heterogeneity problem in these data, without a large gain in the total number of analyzed studies in the meta-analysis. Second, and more importantly, the study sets do not cover the same moderators or tend to cover the moderators in an unbalanced way (e.g., blood studies tend to cover earlier time points as compared to the muscle studies in the acute datasets).

Due to the model selection process, it is difficult to compare the statistics of the meta-regression results across tissues. Nevertheless, we compared the correlation between the effect sizes of the base models of the acute analyses (we ignore the long-term studies in this comparison because there were very few long-term blood studies). This comparison resulted in a mild yet significant correlation (Spearman $\rho=0.1$, $p=1.1e-42$). However, this correlation may be confounded because of the even greater correlation between the I2 scores of the models (Spearman $\rho=0.17$, $p=3.2e-120$). Here are the 2D plots of these statistics:

Due to the risk of confounding and the inability to compare the moderator-based effects directly we decided to omit this comparison from the paper as we fear that it will confuse readers and will blur the message of the paper. Nevertheless, **Supplementary Figure 6** provides the Venn diagram comparing the gene sets identified in the four analyses, providing the reader with an easily interpretable comparison of the differential gene expression overlap.

8. You mention the study by Pillon and heavily criticize their approach. While you may be correct in the shortcomings of their methods, I wonder why you never compared your results with theirs. Just out of curiosity it may be interesting.

The study by Pillon et al. is very interesting and useful as a resource, but as we mention in the paper and the reviewer correctly points out, there were some shortcomings in their methods. They did not follow the guidelines of the meta-analysis/regression literature in several aspects, including how gene heterogeneity was considered, how moderators were handled, and in the way that differential genes were selected. As we explain in the paper, this reduces the credibility of the identified gene sets. We therefore think it would be confusing to the reader and not useful for the community to make a direct comparison based on those shortcomings, and the differences in our approaches.

9. A very large shortcoming of the presented manuscript is that the GEO was searched more than 18 months ago. A current search must be performed if the authors aim to publish their findings in such a high impact journal as “Nature Communications”.

While we of course recognize that 18 months is a rather long time, only 3-4 relevant additional datasets have been added to GEO during this time. For skeletal muscle, this adds ~30 subjects, which is only 4% of the total cohort. While we appreciate this point by the reviewer, we also respectfully want to emphasize the time it takes to properly clean, process, quality control and analyze all data, as well as generate figures and biologically interpret the data. Adding these cohorts would mean redoing the entire analysis. To the reviewers point, the manuscript was also initially submitted to Nature Communications 8 months ago, so 10 months after the initial search. Once 10-15 additional cohorts are available, we think there would be increased value in redoing the analyses and we are hoping additional blood and adipose tissue cohorts will be available soon for us to generate more moderator-specific data in those tissues and allow for more cross-tissue comparisons.

10. In you methods section you spend a lot of time explaining meta-regression to the reader in textbook fashion. This section should be shortened. You also highlight how your AIC approach is more suitable for small sample size. While this is correct, you also only have a relative comparison between models. Could you please provide a R2 for the identified genes?

We thank the reviewer for raising this point. We shortened the text as requested. In addition, we added the R2 scores to **Supplementary Data 5**. However, note that these scores are not available for the base models and models that had a nested cohort|study variance structure (i.e., the rma.mv method from the metafor R package does not output an R2 score in these cases). Thus, only 396 out of the 1063 gene-specific models that satisfied our filters had an R2 score. Moreover, we observed that in the vast majority of the cases the R2 score was high (e.g., 72% of the selected genes in all four analyses had $R^2 > 60\%$ and only 1.6% had $R^2 < 10\%$).

11. Another important drawback of your qPCR analysis in the validation cohort is that you used GAPDH as a control. GAPDH should not be used as a housekeeping gene when performing exercise studies. Exercise training influence the glycolysis of immune cells. Please consider these two manuscripts when identifying appropriate controls:

- https://journals.physiology.org/doi/full/10.1152/physiolgenomics.00025.2005?utm_source=TrendMD&utm_medium=cpc&utm_campaign=Physiological_Genomics_TrendMD_0

- <https://pubmed.ncbi.nlm.nih.gov/12184808/>

We acknowledge that the choice of housekeeping gene is a difficult one that has been extensively discussed in literature. The reason for using GAPDH in this case is that its expression level was

consistent across the human skeletal muscle samples with very low variability (please see attached graph below with mean \pm SD). While there is likely some immune infiltration in skeletal muscle, the majority of the RNA is from skeletal muscle cells. To address the reviewer's concern, we have added one more housekeeping gene, the transcript for the ribosomal protein S18, RPS18 (see specific data for that gene in graph below, again showing the mean \pm SD in the validation cohort). The validation cohort analyses have been updated and are now normalized to the average of the two housekeeping genes.

Minor comments:

12. Why do you capitalize HDL and LDL in the introduction?

We have removed the acronyms from the introduction.

13. Why do you italic "moderators" in the introduction?

We thank the reviewer for this observation, we have removed the italics.

14. you always present the acronym of the genes first followed by the spelled-out name. Please check with the journal formatting guidelines whether this is appropriate

While this is not specified in the journal formatting guidelines, we have read several recently published Nature Communications papers where this order is used for gene names.

15. please spell out BMI at first use

Done

16. please spell out VO2max at first use

Done

17. p16 In 446 it is "cohorts".

We were unable to find this error.

Reviewer #2 (Remarks to the Author):

The authors have addressed all my comments. I am okay with most of the answers. For Figure 3D, I still suggest adding labels for columns in the heatmap.

We greatly appreciate the efforts of the reviewer and have added column labels to Figure 3D according to the suggestion.

Reviewers' Comments:

Reviewer #1:

Remarks to the Author:

The authors have put a lot of effort into revising the manuscript further. Most of my questions were answered sufficiently. I have no further comments.